# *Tet1* regulates epigenetic remodeling of the pericentromeric heterochromatin and chromocenter organization in DNA hypomethylated cells

Yota Hagihara[1], Satoshi Asada[1], Takahiro Maeda[1], Toru Nakano[1,2], Shinpei Yamaguchi[2]*

**1** Graduate School of Frontier Biosciences, Osaka University, Suita, Osaka, Japan, **2** Graduate School of Medicine, Osaka University, Suita, Osaka, Japan

* yamaguchi@patho.med.osaka-u.ac.jp

**Data Availability Statement:** All relevant data are within the manuscript and its Supporting Information files.

## Abstract

Pericentromeric heterochromatin (PCH), the constitutive heterochromatin of pericentromeric regions, plays crucial roles in various cellular events, such as cell division and DNA replication. PCH forms chromocenters in the interphase nucleus, and chromocenters cluster at the prophase of meiosis. Chromocenter clustering has been reported to be critical for the appropriate progression of meiosis. However, the molecular mechanisms underlying chromocenter clustering remain elusive. In this study, we found that global DNA hypomethylation, 5hmC enrichment in PCH, and chromocenter clustering of *Dnmt1*-KO ESCs were similar to those of the female meiotic germ cells. *Tet1* is essential for the deposition of 5hmC and facultative histone marks of H3K27me3 and H2AK119ub at PCH, as well as chromocenter clustering. RING1B, one of the core components of PRC1, is recruited to PCH by TET1, and PRC1 plays a critical role in chromocenter clustering. In addition, the rearrangement of the chromocenter under DNA hypomethylated condition was mediated by liquid-liquid phase separation. Thus, we demonstrated a novel role of *Tet1* in chromocenter rearrangement in DNA hypomethylated cells.

## Author summary

The DAPI-dense chromocenter in the nucleus consists of accumulated pericentromeric heterochromatin. Its number, size, shape, and position are quite different among cell types. One of the striking reorganizations of chromocenter is its clustering, which is observed under DNA hypomethylation conditions, such as in female meiotic germ cells. However, how chromocenters gather and accumulate under the specific epigenetic state remained elusive. Here, we tackled this problem using genetically DNA hypomethylated mouse ESCs and epigenome editing tools. We demonstrated that the DNA demethylation-related factor, TET1 plays a critical role in the chromosome clustering by recruiting Polycomb factors. Forced recruitment of TET protein to the pericentromeric region was

**Funding:** This work was supported by the Takeda Science Foundation to SY, and JSPS KAKENHI (Grant Numbers 16H01217, 16H06223, 19H05754, and 19K06676) to SY. The funders had no role in study design, data collection and analysis, decision to publish, or preparation of the manuscript.

**Competing interests:** The authors have declared that no competing interests exist.

sufficient to induce chromocenter clustering. Thus, we report the novel function of Tet protein in the regulation of large-scale nuclear architecture. These findings will contribute to understanding the molecular mechanism and physiological meaning of chromocenter remodeling, which is frequently observed in developing cells.

## Introduction

The chromosomal region around the centromere, which is named pericentromere, is ~5% in mice. The pericentromeric region of mouse cells is mainly comprised of a 234 bp long AT-rich repetitive sequence, called major satellite repeats. This region forms constitutive heterochromatin harboring high levels of trimethylation of histone H3 lysine 9 (H3K9me3) and DNA methylation at the C-5 position of cytosine (5mC) [1]. At the interphase of the cell cycle, the pericentromeric heterochromatin (PCH) of chromosomes aggregate and form the chromocenter and are visible as 4′6-diamidino-2-phenylindole (DAPI)-dense foci in the nucleus [2].

The organization of chromocenters, including their position, size, and number in the nuclei, are different among individual cell types and are subject to change during their differentiation [3,4]. One of the most striking reorganizations is chromocenter clustering involving the accumulation, and gathering of chromocenters. Chromocenter clustering is observed in germ cells at the meiotic prophase and plays an important role in subsequent meiotic processes, including centromere clustering, homologous chromosome pairing, and synapsis formation [5]. Several factors have been identified to affect chromocenter clustering. For example, MeCP2 and MBD2 localize to the chromocenter and induce clustering during myogenesis, and NANOG is recruited to the chromocenter by SALL1 and controls chromocenter clustering via its transactivation domain in the embryonic stem cells (ESCs) [3,4]. However, previous studies have mainly focused on wild-type ESCs under regular culture conditions and failed to link the clustering mechanism to changes in the epigenetic state.

DNA methylation levels are known to affect the nuclear organization, such as chromocenters [4,6]. Three types of DNA methyltransferases (DNMTs) have been reported to catalyze 5mC in mice: *de novo* DNA methylase DNMT3A and DNMT3B, and maintenance DNA methylase DNMT1 [7]. Loss of the DNMT family of proteins causes global hypomethylation of the CpG sites. In the wild-type ESCs, 82% of CpG sites were methylated, while it was reduced to 18% and 0.4% in the *Dnmt1*-KO ESCs and the *Dnmt1/3a/3b*-TKO (triple knockout of *Dnmt1*, *Dnmt3a*, and *Dnmt3b*) ESCs, respectively [8]. Meanwhile, the TET family comprising TET1, TET2, and TET3 are dioxygenases that convert 5mC to 5-hydroxymethylcytosine (5hmC), 5-formylcytosine (5fC), and 5-carboxylcytosine (5caC) [9]. TET-mediated iterative oxidation is important for the loss of 5mC either through glycosylation by TDG, followed by base excision repair or DNA replication-dependent dilution [9]. In addition, TET proteins have multiple roles, not only in a catalytic activity-dependent but also in an independent manner. For example, TET1 is associated with the SIN3A and NuRD complex as well as pluripotency factors such as NANOG, PRDM14, and LIN28, and regulates the expression of specific genes [10–13]. Considering that 5hmC enrichment in the PCH of meiotic germ cells became undetectable in *Tet1*-KO mice, TET1 could access the PCH locus under these circumstances, but its biological function is poorly understood [14]. Meanwhile, the localization of TET1 is known to overlap with polycomb factors [15,16].

Polycomb group (PcG) repressor proteins are essential for developmental gene regulation and facultative heterochromatin formation [17]. Polycomb repressive complex 1 (PRC1) ubiquitinates lysine 119 of histone H2A (H2AK119ub), which is catalyzed by ubiquitin E3 ligase

RING1A and RING1B (also known as RNF2). PRC2 deposited histone H3 lysine 27 trimethy-lation (H3K27me3) by the core catalytic subunit EZH2. PRC1 and PRC2 act synergistically to suppress the transcription of their target genes; PRC2-mediated H3K27me3 is recognized by PRC1, while H2AK119ub leads to the recruitment of PRC2 [18,19]. Accumulation of PRC1 and PRC2 establishes a long-range interaction between PcG-bound loci and forms a huge pro-tein-DNA complex called the PcG body, which is evident as puncta in the nucleus [20–24].

Meanwhile, liquid-liquid phase separation (LLPS) is characterized by the spontaneous demixing of a homologous solution into two phases by weak forces, such as hydrophobic and electrostatic (pi/pi, cation/pi, etc.) interactions. LLPS contributes to the formation of mem-brane-less organelles and functional drops, including stress granules and PcG bodies [25–27]. Heterochromatin protein α (HP1α), a major chromocenter protein, has been reported to form LLPS-driven droplets *in vitro* [28,29]. However, HP1α foci were resistant to the treatment of 1,6-hexanediol, aliphatic alcohol that disrupts the hydrophobic interaction of LLPS [29]. Therefore, established heterochromatin is believed to comprise mostly immobile compart-ments rather than purely liquid structures.

In this study, we found that ESCs with hypomethylated DNA showed deposition of faculta-tive heterochromatin marks in PCH and chromocenter clustering. Deposition of 5hmC, H2AK119ub, and H3K27me3 in PCH of the DNA hypomethylated ESCs was abrogated by the deletion of *Tet1*. Moreover, the TET1-PRC1 axis was critical for chromocenter clustering in DNA hypomethylated ESCs. We also demonstrated that DNA hypomethylation-dependent chromocenter clustering is driven by LLPS. Thus, our findings revealed a novel function of *Tet1* in the rearrangement of heterochromatin triggered by the loss of DNA methylation.

## Results

### Epigenetic and chromocenter remodeling of the female germ cells and *Dnmt1*-KO ESCs

First, we analyzed HP1γ and H3K9me2, which are catalyzed by G9a H3K9 methyltransferase at the meiotic prophase in female germ cells. PCH forms clusters in the nuclei of both male and female germ cells during meiotic prophase. HP1γ and G9a, both of which are highly enriched in the chromocenter in male germ cells, play a critical role in the clustering of PCH and homologous chromosome pairing in male gonocytes [5]. In contrast, although HP1γ showed specific and consistent localization to PCH, H3K9me2 was depleted in the PCH of female germ cells (S1A and S1B Fig). This distinct enrichment pattern of epigenetic marks sug-gests that the clustering of PCH in male and female germ cells is regulated by different mechanisms.

We then hypothesized that the difference in DNA methylation levels at the meiotic stage would contribute to the sex difference in chromocenter clustering because it affects nuclear architecture [30]. To test this hypothesis, we analyzed the *Dnmt1*-KO ESCs [8]. PCH of the *Dnmt1*-KO ESCs was highly clustered to form large chromocenters in the nucleus, and their number was significantly lower than that of the wild-type ESCs (Fig 1A and 1B). This result was consistent with a previous report showing that DNA hypomethylation causes chromocen-ter clustering [6]. The condensed chromocenter of the *Dnmt1*-KO ESCs retained a strong enrichment of HP1γ, which is similar to meiotic germ cells (Figs 1A and S1A). In contrast, H3K9me2 was not depleted from the chromocenter of both wild-type and *Dnmt1*-KO ESCs (S1C and S1D Fig).

Meanwhile, 5hmC was enriched in the PCH of the *Dnmt1*-KO ESCs, which was also observed in the primordial germ cells (PGCs) during the epigenetic reprogramming phase (S2A Fig) [31]. The relative enrichment score of 5hmC at PCH in the *Dnmt1*-KO ESCs was

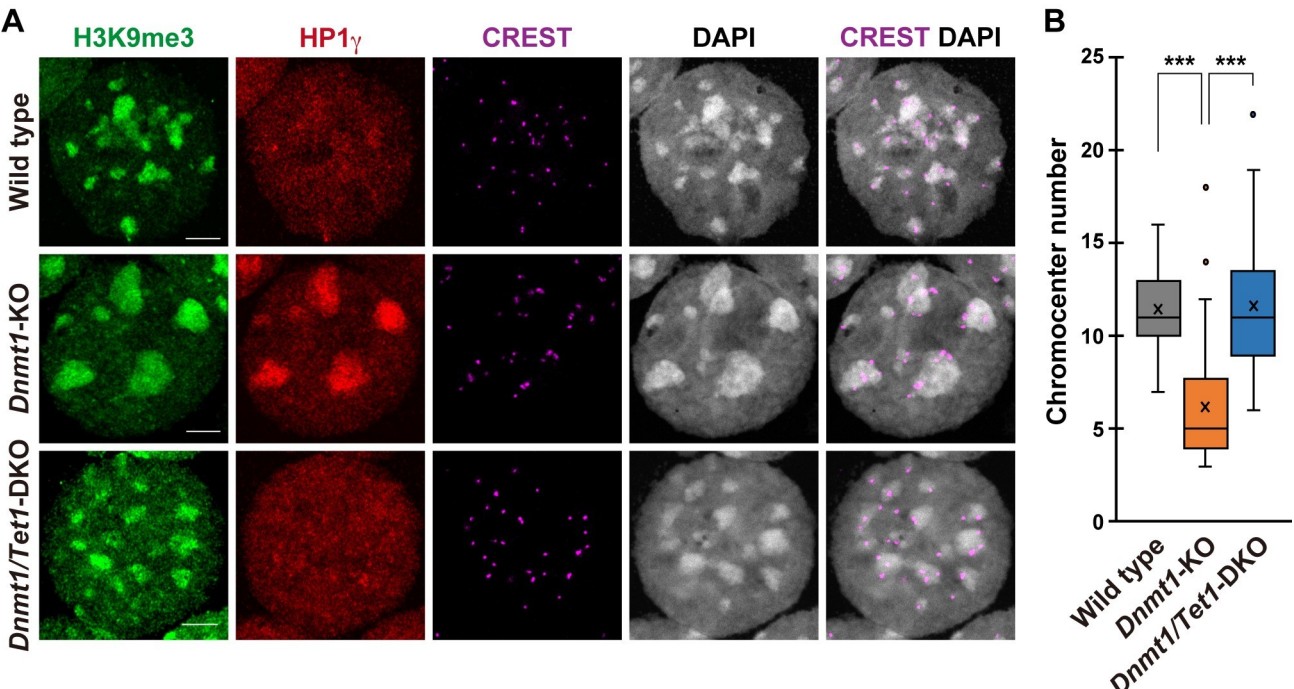

**Fig 1. *Tet1*-dependent chromocenter clustering in the DNA hypomethylated ESCs. (A)**: Representative immunostaining images of the surface-spread nuclei of the wild-type, *Dnmt1*-KO and *Dnmt1/Tet1*-DKO ESCs. Scale bar, 5 μm. **(B)**: Boxplot showing the number of distinct chromocenters in the nuclei of ESCs, related to (A). In boxplots, the enter lines show the medians; box limits indicate the 25th and 75th percentiles, and whiskers extend 1.5 times the interquartile range from the 25th and 75th percentiles. Actual values are indicated by circles. Wild-type, n = 55; *Dnmt1*-KO, n = 64; *Dnmt1/Tet1*-DKO, n = 64. *P* values were calculated using the Mann-Whitney U-test. ***P < 0.001.

about two times higher than that of the wild-type ESCs (1.21 vs. 0.62) (S2B Fig). These results demonstrated that the *Dnmt1*-KO ESCs exhibited features similar to those of female germ cells, such as global DNA hypomethylation, PCH-5hmC, and clustered chromocenters, to some extent.

### The critical function of *Tet1* in 5hmC deposition at PCH and chromocenter clustering

Considering that TET1 is critical for the acquisition of PCH-5hmC in PGCs and is highly expressed in ESCs, we focused on the function of TET1 [14]. Immunocytochemical analysis demonstrated that TET1 was slightly but significantly enriched in the chromocenter of the *Dnmt1*-KO ESCs compared to the wild-type ESCs (S2D and S2E Fig). Taken together, we hypothesized that TET1 was responsible for the gain of 5hmC at PCH and plays a role in chromocenter clustering in DNA hypomethylated cells. To validate this hypothesis, we knocked out *Tet1* in the *Dnmt1*-KO ESCs by CRISPR-mediated mutagenesis, and the depletion of TET1 protein was confirmed by immunocytochemistry and Western blotting analysis (S2D and S2F Fig) [32].

Enrichment of 5hmC in PCH of the *Dnmt1/Tet1*-DKO ESCs was significantly decreased compared to the *Dnmt1*-KO ESCs and was comparable to that of the wild-type ESCs (0.54 vs. 0.62) (S2B Fig). In contrast, the 5mC enrichment in PCH of the *Dnmt1/Tet1*-DKO ESCs was about twice that of the wild-type ESCs (16.4 vs. 8.1) (S2C Fig). These results showed that TET1 actively oxidizes 5mC to 5hmC at PCH in the *Dnmt1*-KO ESCs. Remarkably, *Tet1* depletion resulted in a dispersed pattern of chromocenters and an increase in the number comparable to

that of the wild-type ESCs (Fig 1A and 1B). These results demonstrate that *Tet1* is critical for chromocenter clustering in DNA hypomethylated ESCs.

To investigate whether the chromocenter clustering was mediated by the TET1 oxidative products, we analyzed *Dnmt1/3a/3b*-TKO ESCs, in which 5mC nor 5hmC was detectable [8,33]. Chromocenters of the *Dnmt1/3a/3b*-TKO ESCs were similar to those of the *Dnmt1*-KO ESCs (Figs 1A, S2G and S2H). In contrast, the chromocenters of the *Dnmt1/3a/3b/Tet1*-QKO (quarto knock out of *Dnmt1*, *Dnmt3a*, *Dnmt3b*, and *Tet1*) ESCs were dispersed in the nucleus and their number was significantly higher than that of the *Dnmt1/3a/3b*-TKO ESCs (S2H Fig). The number of chromocenters in *Dnmt1/3a/3b/Tet1*-QKO ESCs was essentially the same as that of the wild-type and the *Dnmt1/Tet1*-DKO ESCs (S2G and S2H Fig). These results demonstrate that *Tet1*-mediated oxidative products of 5mC, such as 5hmC, 5fC, and 5caC are not necessary for chromocenter clustering.

## Induction of chromocenter clustering by the forced recruitment of TET1

To address whether TET1 localization to PCH is sufficient to induce chromocenter clustering, we carried out an experiment recruiting TET1 to PCH using the epigenome editing method [34]. This dCas9-SunTag system recruits the TET1 catalytic domain (CD) to single-guide RNA (sgRNA) targeted regions through anti-GCN4 scFv domains by tethering dCas9-fused GCN4 (Fig 2A) [34]. We designed a sgRNA targeting major satellite repeats to recruit TET1-CD to PCH. A whole cassette without sgRNA and scFv-only cassettes were used as negative controls (Fig 2B).

The sgRNA-mediated TET1 recruitment to the PCH was verified by the specific enrichment of GFP (Figs 2C and S3A). As shown in S3B and S3C Fig, 5hmC was strongly enriched in PCH of the TET1-CD recruited ESCs but was absent in PCHs of the negative control and TET1 catalytic dead mutant (CD-Mut)-recruited ESCs. These results indicated that TET1-CD was successfully recruited to PCH in a sgRNA-dependent manner. In the TET1-forced recruited ESCs, chromocenters were clustered, and their numbers were significantly decreased in not only TET1-CD recruited ESCs but also TET1-CD-Mut recruited ESCs (Fig 2C and 2D). These results clearly showed that the recruitment of TET1-CD to PCH was sufficient to induce chromocenter clustering in a catalytic activity-independent manner.

## Contribution of PRCs to chromocenter clustering under the DNA hypomethylated state

To address the molecular mechanism of *Tet1*-mediated chromocenter clustering in the DNA hypomethylated ESCs, first, we analyzed *Nanog*, which was reported to regulate chromocenter clustering in ESCs [3]. Unfortunately, however obvious changes in neither expression level nor localization of NANOG in the *Dnmt1*-KO ESCs and the *Dnmt1/Tet1*-DKO ESCs was not detected (S4A–S4C Fig).

Then, we hypothesized that Polycomb factors would be the core of chromocenter clustering mediated by *Tet1* because of their aggregate-forming ability and co-localization with TET1 [15]. TET1 distribution overlaps with PRC2 and H3K27me3 in mouse ESCs [15]. The loss of DNA methylation has been demonstrated to induce PRC2 and H3K27me3 enrichment in PCH [19,35]. Consistently, H3K27me3 at PCH, which was depleted in the wild-type ESCs, was increased in the *Dnmt1*-KO ESCs (S5A and S5B Fig). In contrast, the immunocytochemical analysis showed that the level of H3K27me3 of PCH in the *Dnmt1/Tet1*-DKO ESCs was lower than that of the *Dnmt1*-KO ESCs and was similar to that of the wild-type ESCs (S5A and S5B Fig). *Tet1*-dependent H3K27me3 enrichment in the major satellite repeats of *Dnmt1*-KO ESCs was confirmed by chromatin immunoprecipitation (ChIP)-qPCR analysis (S5C Fig).

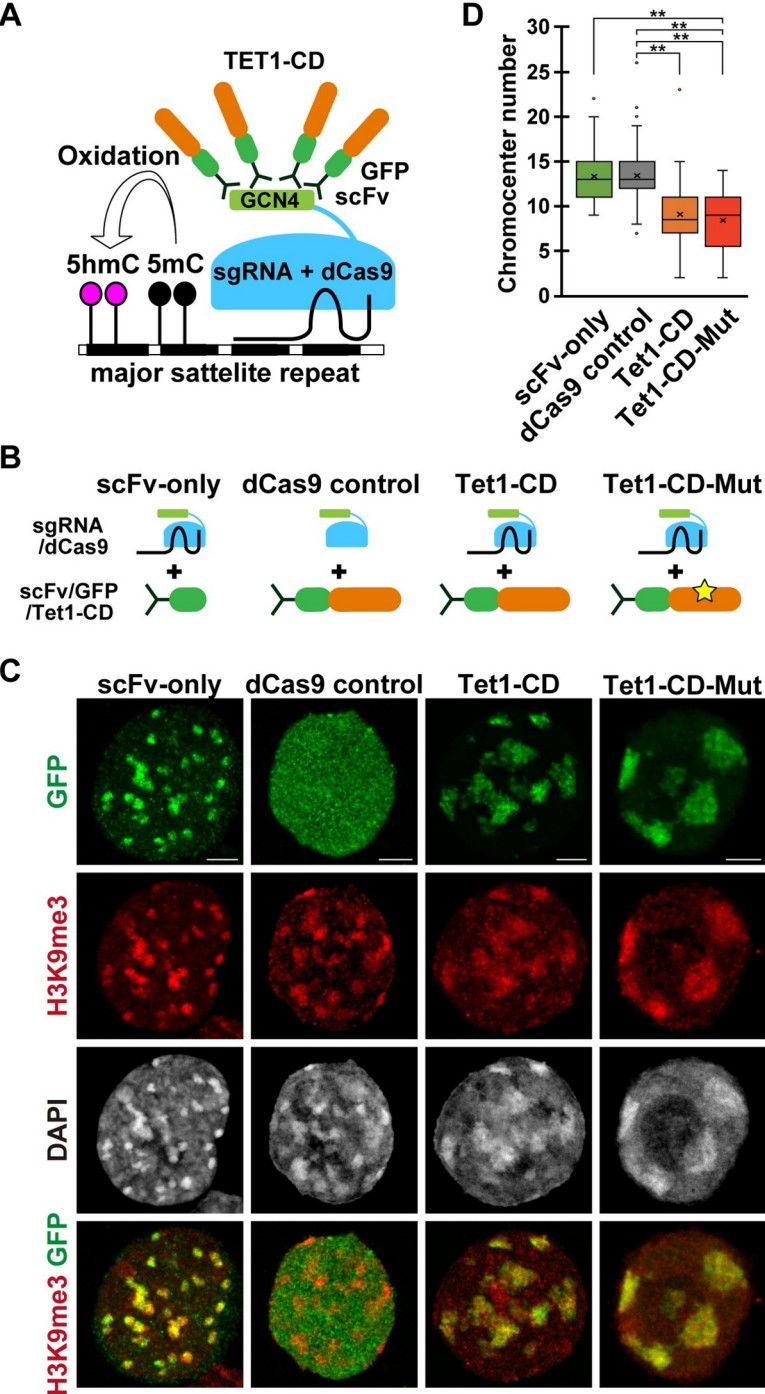

**Fig 2. Forced recruitment of TET1-CD to PCH and chromocenter clustering.** **(A)**: Schematic illustration of the experimental design. GCN4 was conjugated with catalytically-dead Cas9 (dCas9), which was recruited to PCH by the sgRNA targeting major satellite repeats, and GCN4 was targeted by scFv conjugated with TET1-CD. **(B)**: Schematic illustration of experimental conditions. **(C)**: Representative immunostaining images of the surface-spread nuclei of the ESCs expressing individual cassette. Scale bar, 5 μm. **(D)**: Boxplot showing the number of distinct chromocenters in the nuclei of ESCs expressing individual cassette, related to (C). scFv-only, n = 34; dCas9 control, n = 48; TET1-CD, n = 44; TET1-CD-Mut, n = 41. *P* values were calculated using the Mann-Whitney U-test. \*\*\**P* < 0.001.

Consistently, EZH2 enrichment to PCH was increased in the *Dnmt1*-KO ESCs and decreased in the *Dnmt1/Tet1*-DKO ESCs (S5D and S5E Fig). These results suggest that *Tet1* plays a role in recruiting PRC2 to PCH under DNA hypomethylated condition.

We then depleted *Ezh2*, which is the core subunit of PRC2, in the *Dnmt1*-KO ESCs to test whether PRC2 was critical for chromocenter clustering (S6A Fig). Neither EZH2 nor H3K27me3 were detectable by immunostaining in two independent *Dnmt1/Ezh2*-DKO ESC clones (S6B Fig). Enrichment of TET1 and H3K9me3 in PCH in two clones of *Dnmt1/Ezh2*-DKO ESCs was similar to the control (S6C–S6E Fig). However, the pattern of chromocenter clustering was quite different between these two clones; one clone showed no significant increase, whereas the other clone showed a slight but significant increase (S6F Fig). HP1γ enrichment at chromocenter was reduced in both clones (S6G Fig). Even if there are some, PRC2 would have only a limited contribution to chromocenter clustering under the DNA hypomethylated state.

H2AK119ub, which was catalyzed by PRC1, was enriched in the PCH of *Dnmt1*-KO ESCs as previously reported (Fig 3A and 3B) [19]. Both immunostaining and ChIP-qPCR analysis showed that H2AK119ub enrichment in the PCH of *Dnmt1*-KO ESCs depended on *Tet1* (Fig 3A–3C). RING1B, the central component of PRC1, was also increased in the chromocenter of *Dnmt1*-KO ESCs, while it was decreased in the *Dnmt1/Tet1*-DKO ESCs (S7A and S7B Fig). Co-immunoprecipitation assay demonstrated that the TET1-CD formed a complex with RING1B, suggesting that TET1 directly associates and recruit PRC1 to the target (S7C Fig). This hypothesis was supported by the increased RING1B enrichment at the chromocenter of TET1-CD- and TET1-CD-Mut-recruited ESCs (S7D and S7E Fig). In contrast, both RING1B and H2AK119ub at PCH of DNA hypomethylated ESCs exhibited little effects by the depletion of *Ezh2* (S8A–S8D Fig). These data showed that TET1, but not PRC2 plays a critical role in the recruitment of PRC1 to the chromocenter and regulates the epigenetic state of PCH under DNA hypomethylated condition.

To test whether PRC1 is involved in chromocenter clustering, we targeted *Ring1A* and *Ring1B*, which are core components of PRC1 [36]. It was almost impossible to establish stable *Ring1A/1B*-DKO ESC lines because of the proliferation defect and differentiated phenotype of *Ring1A/1B*-depleted ESCs [37]. We then analyzed the *Dnmt1/Ring1A/1B*-TKO ESCs shortly after the introduction of the CRISPR cassette encoding Cas9 and sgRNAs (S9A Fig).

ESCs with depleted *Ring1A/1B* should have been detectable as H2AK119ub-negative cells since H2AK119ub synthesis completely depends on RING1A/1B [37]. Although H2AK119ub-negative cells were not detectable in the control wild-type or *Dnmt1*-KO cells, these cells appeared 4 d after the transfection of the CRISPR cassette (Fig 3D). These H2AK119ub-negative cells showed a dispersed pattern of chromocenter, and its number was comparable to that of the wild-type ESCs (Fig 3D and 3E). Notably, HP1γ enrichment at PCH did not change in the *Dnmt1/Ring1A/1B*-TKO ESCs, suggesting that HP1γ was insufficient for chromocenter clustering under DNA hypomethylated condition. Considering that the enrichment of TET1 at PCH was slightly but significantly affected in the *Dnmt1/Ring1A/1B*-TKO ESCs, PRC1 complex would play some roles to stabilize TET1 at the PCH (S9B and S9C Fig). These results demonstrate that *Tet1*-dependent localization of PRC1 to PCH is one of the molecular bases of chromocenter clustering in DNA hypomethylated ESCs.

## Liquid-liquid Phase separation (LLPS) contributes to chromocenter clustering in the DNA hypomethylated ESCs

Based on the hypothesis that LLPS drives *Tet1*-dependent large chromocenter clustering in DNA hypomethylated ESCs, we carried out real-time monitoring of the dynamic behavior of

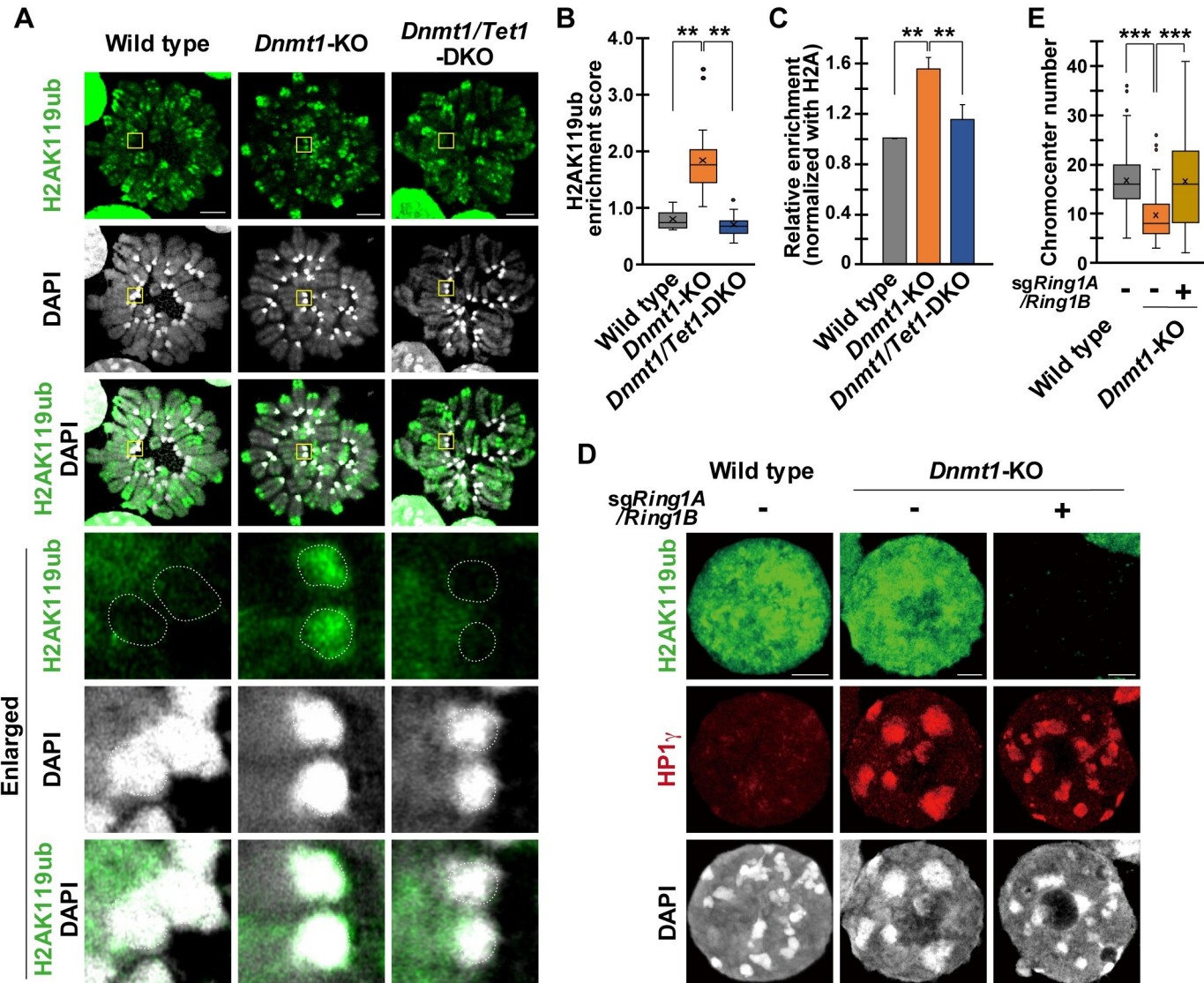

**Fig 3. The critical role of PRC1 in the chromosome clustering in the DNA hypomethylated ESCs.** (**A**): Representative immunostaining images of the surface-spread nuclei of the wild-type, *Dnmt1*-KO and *Dnmt1/Tet1*-DKO ESCs. The yellow squares indicate the enlarged areas shown on the bottom. The white dashed circles indicate the PCH. (**B**): Quantification of immunostaining of H2AK119ub, related to (A). The enrichment score is calculated by the relative signal intensity of PCH to the entire chromosome. An average of 3 chromosomes was calculated for each cell. n = 20 per cell line. (**C**): ChIP-qPCR for H2AK119ub at the major satellite repeats in the wild-type, *Dnmt1*-KO and *Dnmt1/Tet1*-DKO ESCs. Each bar represents relative enrichment to wild-type after normalization against H2A ChIP. n = 3 per cell line. (**D**): Representative immunostaining images of surface-spread nuclei of the wild-type, *Dnmt1*-KO and *Dnmt1/Ring1A/1B*-TKO ESCs. Cells were analyzed 4 d after the transfection of the CRISPR cassette. (**E**): Boxplot showing the number of distinct chromocenters in the nuclei of ESCs, related to (D). Wild-type, n = 97; *Dnmt1*-KO, n = 75; *Dnmt1/Ring1A/1B*-TKO, n = 64. *P* values were calculated using the Mann-Whitney U-test. ***$P < 0.001$; **$P < 0.01$. Scale bar, 5 μm.

the chromocenters. For this purpose, we established ESC lines expressing humanized Kusabira-Orange-1-tagged HP1γ (hKO1-HP1γ), which was specifically localized to PCH [38]. Time-lapse analysis of the *Dnmt1*-KO ESCs with the hKO1-HP1γ reporter revealed that their chromocenters underwent fusion and round up and produced larger chromocenters (Fig 4A and S1 Movie). This dynamic change in the nucleus indicated that the chromocenter of DNA hypomethylated cells possessed the property of liquid-like condensates.

LLPS-driven condensation can be disrupted by 1,6-hexanediol (HD) treatment *in vivo* [27,39]. We treated ESCs with 1,6-HD and monitored the immediate response to

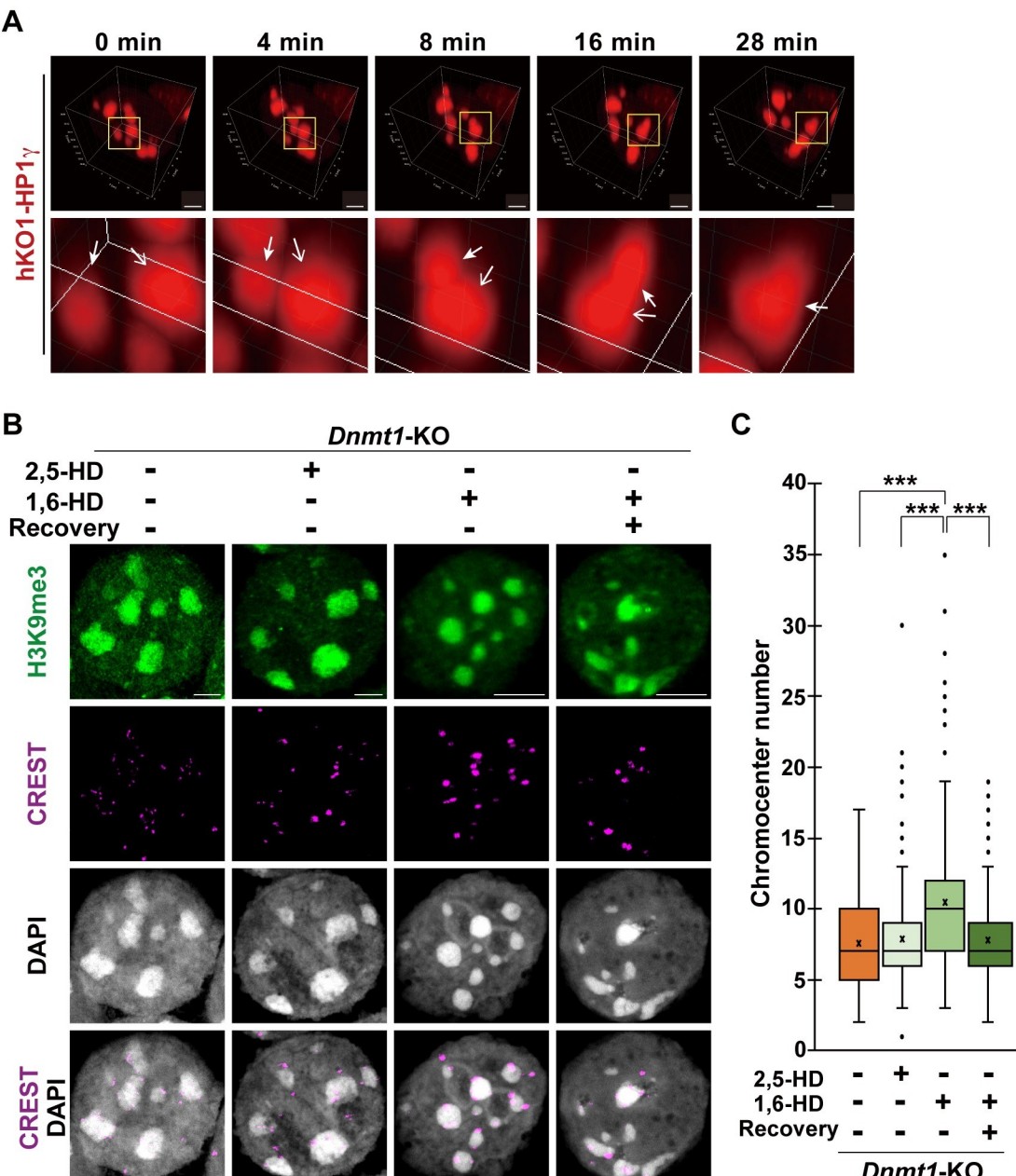

**Fig 4. Liquid-liquid phase separation (LLPS) drives chromocenter clustering in the DNA hypomethylated cells. (A)**: Series of time-lapse 3D reconstruction images of the *Dnmt1*-KO ESCs expressing hKO1-HP1γ. Yellow squares indicate the region highlighted at the bottom panel. White arrows indicate the chromocenters undergoing the fusion. Images were captured and cropped from S1 Movie. Scale bar, 3 μm. **(B)**: Representative immunostaining images of surface-spread nuclei of the *Dnmt1*-KO ESCs with or without 1,6-hexanediol (HD) treatment. About 10% of 2,5- or 1,6-HD were treated for 1 min. For the recovery experiment, cells were incubated in an ESC culture medium for 1 h. Scale bar, 5 μm. **(C)**: Boxplot showing the number of distinct chromocenters in the nuclei of the *Dnmt1*-ESCs in the individual condition, related to (B). No treatment, n = 71; 2,5-HD treatment, n = 222; 1,6-HD treatment, n = 217; 1,6-HD treatment with recovery culture, n = 199. *P* values were calculated using the Mann-Whitney U-test. ***$P < 0.001$.

chromocenter clustering. The number and shape of the chromocenter of the wild-type ESCs were not affected by 1-min treatment with 1,6-HD, when compared to 2,5-HD, which is an isomer of 1,6-HD and less effective in disrupting LLPS (S10A and S10B Fig). This observation

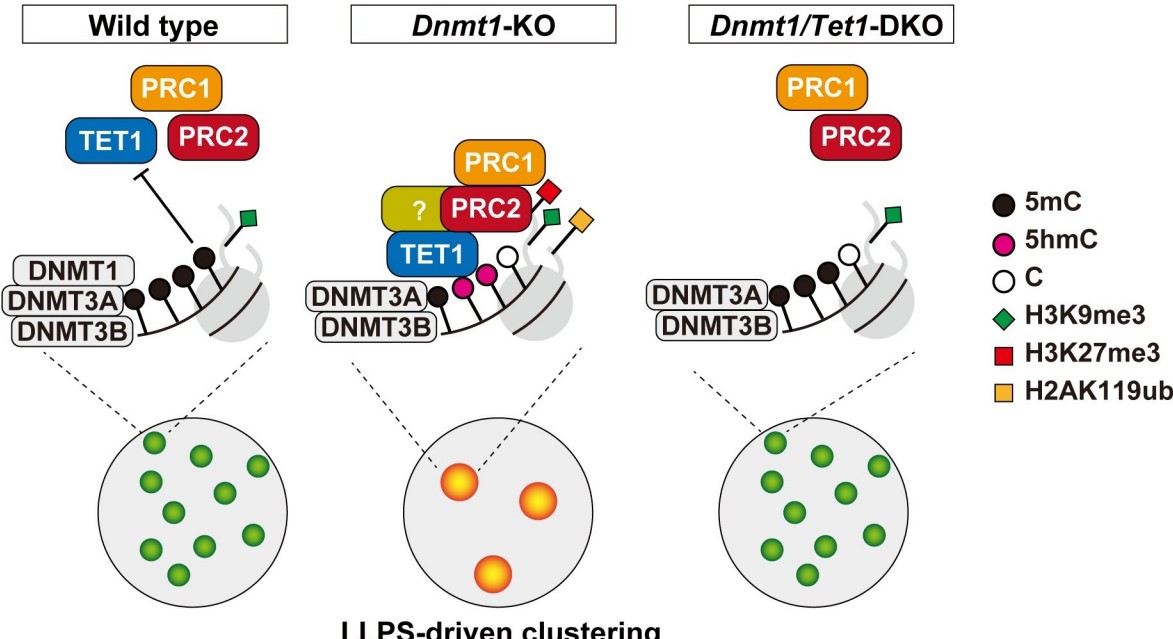

**Fig 5. A model of *Tet1*-mediated chromocenter clustering in DNA hypomethylated cells.** CpG sites in the PCH were highly methylated by DNMTs and not accessible by TET1 and PcG proteins in the wild-type ESCs (Left panel). DNA hypomethylated condition allows TET1 to bind PCH region and TET1 recruits PcG proteins in the *Dnmt1*-KO ESCs. Thus, 5hmC, H2AK119ub, and H3K27me3 are enriched in PCH, and the chromocenters are clustered through the LLPS mechanism (Middle panel). PcG proteins are not recruited to PCH and chromocenters remain dispersed in the *Dnmt1/Tet1*-KO ESCs (Right panel).

was consistent with a previous study that showed that 1,6-HD had limited effects on the established heterochromatin [29]. In contrast, the number of chromocenters was significantly increased by 1,6-HD in the *Dnmt1*-KO ESCs (Fig 4B and 4C). Furthermore, the number of chromocenters was reduced to a level comparable to that of the untreated or 2,5-HD-treated *Dnmt1*-KO ESCs after removal of 1,6-HD (Fig 4B and 4C). Taken together, we conclude that *Tet1*-dependent chromocenter clustering in DNA hypomethylated ESCs is regulated by LLPS. However, the chromocenter number of the *Dnmt1*-KO ESCs was still less than that of the wild-type ESCs, even with 1,6-HD treatment (Figs 4C and S10B). It is conceivable that some other physiological interactions, which are resistant to 1,6-HD, play a role in chromocenter clustering in *Dnmt1*-KO ESCs.

Based on the above results, we propose that *Tet1* induces epigenetic reprogramming of PCH under DNA hypomethylated condition, namely, deposition of 5hmC and facultative heterochromatin marks of H2AK119ub and H3K27me3 (Fig 5). Tet1-mediated reprogramming further led to chromocenter clustering through LLPS (Fig 5). This TET1-PRC1 axis is likely the molecular basis of chromocenter reorganization under the DNA hypomethylated condition.

## Discussion

DNA in the PCH region is usually highly methylated and is presumably not targeted by TET proteins, as evident through undetectable 5hmC [40]. On the contrary, enrichment of 5hmC in PCH was obvious in the *Dnmt1*-KO ESCs and wild-type PGCs during epigenetic reprogramming, both of which possessed globally DNA hypomethylated states (S2A Fig) [31]. Although *Tet1* and *Tet2* are highly expressed in ESCs and PGCs, the depletion of *Tet1* resulted

in complete abrogation of PCH-5hmC in the *Dnmt1*-KO ESCs and PGCs (S2A Fig) [14]. The CXXC domain, which can bind to unmethylated DNA, is present in TET1 but not in TET2, and this difference may account for the difference in PCH targeting [41]. On the other hand, chromocenter clustering occurred in the *Dnmt1/3a/3b*-TKO ESCs and forced recruitment of TET1-CD-Mut to PCH induced its clustering. Therefore, PCH-5hmC was suggested to be a byproduct and not crucial for the chromocenter clustering.

A gain of facultative heterochromatin marks, such as H3K27me3 and H2AK119ub, at PCH, indicates that epigenetic heterochromatin remodeling occurred under the DNA hypomethylated state. We demonstrated that *Tet1* is critical for this DNA hypomethylation-linked heterochromatin remodeling. A total of 97.2% of PRC2-binding sites overlapped with TET1 binding sites and 72.2% of PRC2 binding was affected by the depletion of *Tet1* [16]. Consistent with these studies, we found that H3K27me3 levels at the PCH were decreased in the *Dnmt1/Tet1*-DKO ESCs (S5A–S5C Fig).

We also found that PRC1 recruitment to PCH in DNA hypomethylated ESCs depends on *Tet1* (Figs 3A, 3B, S7A and S7B). Given that PRC1 binds to H3K27me3 through CBX7 in ESCs, the localization of PRC1 to PCH would be mediated by PRC2 [42]. However, enrichment of H2AK119ub in PCH was more intense than that of H3K27me3, PRC2 depletion showed minor effect on enrichment of H2AK119ub and RING1B, and the phenotype of the chromocenter clustering in the *Ring1A/1B*-DKO ESCs was more obvious than that of the *Ezh2*-KO ESCs. Furthermore, forced recruitment of the TET1 catalytic domain, which is associated with PRC1, to PCH induced the chromocenter clustering. Thus, we propose a model in which PRC1 is first recruited to PCH by TET1, followed by the binding of PRC2 to PRC1-mediated H2AK119ub. This model was supported by a study that forced recruitment of MBD-EZH2 to the chromocenter did not lead to H2AK119ub enrichment, but the recruitment of MBD-RING1B induced H3K27me3 enrichment [19]. It has also been reported that 90% of RING1B targets in ESCs correspond to bivalent promoters, which are the main targets of TET1 [9,43].

The PcG body is associated with the enrichment of facultative heterochromatin marks, H2AK119ub and H3K27me3 [22]. The puncta containing RING1B show a liquid-like feature of fusion and fission [21]. CBX2, the canonical PRC1 component, mediates PcG body formation in an LLPS-dependent manner [26,27]. Both clustered chromocenters in the DNA hypomethylated cells and PcG body exhibited PRC1 inclusion and LLPS characteristics (Figs 4B, 4C, S7A and S7B). Therefore, it is reasonable to consider that chromosome clustering in DNA hypomethylated cells relies on the biochemical characteristics of PcG proteins, which have the potential to aggregate into large foci.

PCH is represented by the constitutive heterochromatin marks such as H3K9me3 and HP1. H3K9me3 and HP1 have been known to regulate each other at PCH. For example, *Suv39h1/h2*, an enzyme responsible for the H3K9 methylation at the constitutive heterochromatin, plays a pivotal role in HP1 deposition, and HP1 depletion affects the accumulation of H3K9me3 at PCH [35,44]. However, we found only the HP1γ enrichment, but no alteration of H3K9me3 in the PCH of *Dnmt1*-KO ESC. Thus, the HP1γ enrichment in the PCH of the DNA hypomethylated ESCs was quite likely to be independent of H3K9me3 and was regulated by TET1 (Fig 1A). In contrast, the enrichment of HP1γ in PCH was not affected in the meiotic germ cells of *Tet1*-KO mice [14]. These results suggest that DNA methylation, *Tet1*, *Suv39h1/h2*, and PRC1/2 are intricately regulated among them and that the epigenetic state of PCH is regulated by distinct mechanisms in different cell types.

Chromocenter clustering is believed to lead to spatial reorganization of the nucleus and affect the microenvironment for gene silencing. Pericentromeric regions are reported to associate with other genomic regions and form a large domain called the pericentromere-associate

domain (PAD) [45]. PAD patterns vary in different cell types and dynamically change during differentiation, unlike topologically associating domains (TADs), which are less variable among different cell types [45,46]. Forced recruitment of specific genes to PCH using chromo-domain of HP1β resulted in H3K9me3 deposition and transcriptional silencing of that gene. Therefore, chromocenter reorganization, accompanied by chromocenter clustering and its enlargement, may change PAD and provide more space to repress other genomic loci.

Heterochromatin remodeling accompanied by ectopic PRC1 localization to PCH reduces H2AK119ub enrichment in other genomic loci [47]. The high copy number of satellite repeats has been proposed to act as a molecular "sponge", which can harbor epigenetic factors abundantly and affect the enrichment of the proteins in the genome [47]. Considering that ectopic localization of both PRC1 and PRC2 to the chromocenter was observed, re-localization of PRC1 and PRC2 and de-repression in other genomic regions might occur in the DNA hypo-methylated ESCs.

Chromocenter clustering occurs at the meiotic prophase and is important for the progression of meiosis in germ cells [5]. Male germ cells enter meiosis under the DNA hypermethylated state, whereas female germ cells initiate meiosis under the DNA hypomethylated state [48]. Since the DNA methylation status of female meiotic germ cells is similar to that of *Dnmt1*-KO ESCs, *Tet1* potentially plays a role in chromocenter clustering during meiosis in female germ cells.

In summary, we identified a novel role of *Tet1* in the epigenetic remodeling of PCH and chromocenter clustering through PRC1 recruitment in DNA hypomethylated ESCs. Previous studies on PCH have mainly focused on the factors related to constitutive heterochromatin, such as SUV39H1/2 or HP1 [49,50]. However, our findings demonstrated that TET1 and PRC1, both of which are facultative heterochromatin factors, control the spatial organization of the chromocenter. Global DNA hypomethylation and PRC1 localization to the chromocenter have been reported in pre-implantation embryos and cancer cells [47,51]. TET family proteins would also regulate epigenetic remodeling of the PCH of these cells. Thus, our findings shed light on the field of chromocenter organization.

## Materials and methods

### ESCs culture

Wild type (J1 strain) and KO ESCs were maintained in Glasgow modified Eagle's medium (GMEM) containing high glucose supplemented with 10% fetal calf serum, 100 U/mL mouse LIF, 2-mercaptoethanol, L-glutamine, penicillin/streptomycin, 3 μM CHIR99021 (Chems-cene), and 1 μM PD0325901 (CAYMAN) on gelatin-coated culture plates. *Dnmt1*-KO and *Dnmt1/3a/3b*-TKO ESCs were provided by Dr. Okano (Kumamoto University, Japan) [8].

### Generation of knockout cell lines

All knockout ESC lines were established using the CRISPR/Cas9 system. ESCs were transfected with a mixture of px330-puro vectors containing specific sgRNAs targeting each gene (S1 Table). Successful depletion of target genes was verified using genotyping PCR, Western blotting, and immunocytochemical analyses.

As for the depletion of *Ring1A/1B*, *Dnmt1*-KO ESCs was transfected with a mixture of px330-Puro vector containing sgRNA targeting *Ring1A/1B* (S1 Table). For selection, ESCs were cultured with 1 μg/mL puromycin for 2 d. Subsequently, the cells were cultured for recovery without puromycin for 1 d. The cells were then harvested and used for surface spread preparation.

## Epigenome editing experiments

The fragment for sgRNA targeting the major satellite repeat (S1 Table) was inserted into the pPlatTET-gRNA2 plasmid (Addgene #82559). The scFv-only vector was generated by removing the TET1-CD domain from this plasmid. The TET1-CD-Mut vector contains nucleotide mutations corresponding to two amino acids critical for Tet1 catalytic activity (H1671Y and D1673A of TET1 protein) [52]. These plasmids were transfected to ESCs using Lipofectamine 2000 (Invitrogen, 11668019). Followed by the culture for 2 d, the cells were fixed for the immunofluorescent analysis, or harvested for surface spread preparation. GFP-positive cells were randomly picked for the following analysis.

## Surface spread and immunocytochemistry

ESCs and gonadal cells were dissociated and placed on a glass slide dipped in a fixation solution of 1% paraformaldehyde (PFA) in distilled water (DW) (pH 9.2) containing 0.15% Triton X-100 and 3 mM dithiothreitol. Slides were then incubated overnight at 4˚C for a fixation. The slides were then washed in 0.4% Photoflo (Kodak) in DW and dried at room temperature. The samples were washed with 0.1% Triton X-100/PBS (PBS with Tween 20; PBST) and treated with 0.5% Triton X-100 for 20 min. After washing with PBST, the samples were incubated with 3% BSA and 2% goat serum in PBST (blocking buffer). Samples were then incubated with primary antibodies diluted in blocking buffer, washed, and incubated with appropriate secondary antibodies. Immunofluorescence was visualized using an LSM880 (Carl Zeiss). The enrichment score of each chromosome was calculated by the relative signal intensity of PCH to the entire chromosome using ZEN software (Zeiss). The enrichment score of each cell was determined by an average the enrichment scores of three chromosomes.

## Immunocytochemistry using culture slide

For immunostaining against RING1B, TET1, EZH2, and NANOG, ESCs were plated in a culture slide (Corning, #354118) and cultured for 1 d before immunostaining. ESCs were fixed with 4% PFA for 5 min at room temperature, followed by permeabilization with 0.5% Triton X-100/PBS for 20 min. They were then incubated with the primary antibodies diluted in the blocking buffer at 4˚C overnight. The samples were washed with PBST and incubated with the secondary antibodies. Immunofluorescence was visualized using an LSM880 (Carl Zeiss). Relative signal intensity of the chromocenter was determined by dividing the average signal intensity of three to six chromocenters by that of whole nuclear area using ZEN software (Zeiss).

## Antibodies

Primary antibodies used in this study include rabbit anti-5hmC (Active Motif, #39769), mouse anti-5mC (Active Motif, #39649), rabbit anti-NANOG (Abcam, ab80892), rabbit anti-H3K9me2 (Abcam, ab1220), rabbit anti-H3K9me3 (Abcam, ab8898), mouse anti-H3K27me3 (Wako, MABI0323), rabbit anti-H2AK119ub (Cell signaling, #8240), mouse anti-γH2AX (Sigma-Aldrich, 07–164), mouse anti-SCP3 (Santa Cruz, sc-74569), rabbit anti-SCP3 (Novus, NB-300-232), mouse anti-HP1γ (Invitrogen, MA3-054), human anti-Centromere Protein (CREST) (Antibodies Incorporated, 15–235), rabbit anti-TET1 (Millipore, 09–872), mouse anti-TET1 (Active Motif, 91172), rabbit anti-EZH2 (Cell signaling, #5246), rabbit anti-GFP antibody (Invitrogen, A11122), and rabbit anti-RING1B (Cell signaling, #5694). The secondary antibodies used in this study included Alexa Fluor 488 goat anti-mouse IgG, Alexa Fluor

488 goat anti-rabbit IgG, Alexa Fluor 568 goat anti-mouse IgG, Alexa Fluor 568 goat anti-rabbit IgG, and Alexa Fluor 647 goat anti-human IgG (Invitrogen).

## ChIP-qPCR

The number of ESCs were adjusted, washed with PBS, and cross-linked with 1% formaldehyde for 5 min, and quenched with 125 mM of Glycine. After the wash with PBS, cells were resuspended in lysis buffer (50 mM HEPES (pH 7.9), 140 mM NaCl, 1 mM EDTA, 0.5% NP-40, 0.25% TritonX-100, 10% Glycerol). After 5 min incubation, the cells were washed twice in wash buffer (10 mM Tris (pH 8.1), 200 mM NaCl, 1 mM EDTA, 0.5 mM EGTA). The cells were resuspended in shearing buffer (10 mM Tris (pH 8.1), 1 mM EDTA, 0.1% SDS) and sonicated using S220 focused-ultrasonicator (Covaris) for 30 min. Immunoprecipitations were performed with the diluted cell lysis corresponding to 1000 cells and antibody-linked Dynabeads (Invitrogen, 10003D) in IP buffer (16.7 mM Tris (pH 8.1), 167 mM NaCl, 1.2 mM EDTA, 1.1% TritonX-100, 0.01% SDS) for 2 h. After washing in Low-salt buffer (20 mM Tris (pH 8.0), 150 mM NaCl, 2 mM EDTA, 1% TritonX-100, 0.01% SDS), High-salt buffer (20 mM Tris (pH 8.0), 500 mM NaCl, 2 mM EDTA, 1% TritonX-100, 0.01% SDS), LiCl buffer (10 mM Tris (pH 8.1), 250 mM LiCl, 1 mM EDTA, 1% NP-40, 1% Sodium deoxycholate), and TE, the chromatin was reverse cross-linked in elution buffer (10 mM Tris (pH 8.0), 300 mM NaCl, 1 mM EDTA) supplemented with Proteinase K (Nakarai, 29442–14) at 58°C for overnight. DNA was purified with QIAquick PCR purification kit (QIAGEN) and quantified by Thunderbird qPCR Mix (TOYOBO, QPK-201) with the major satellite specific primers (5'-GAC GACTTGAAAAATGACGAAATC-3' and 5'-CATATTCCAGGTCCTTCAGTGTGC-3'). Antibodies used were mouse anti-H3K27me3 (Wako, MABI0323), mouse anti-H3 (Cosmo Bio, #MCA-MABI0001-100), rabbit anti-H2AK119ub (Cell signaling, #8240), and rabbit anti-H2A (Active Motif, # 39945).

## Co-immunoprecipitation assay

A mixture of GFP-Ring1B and FLAG-TET1-CD plasmids were transfected to 293T cells and cultured for 2 d. The cells were harvested and resuspended in whole cell lysis buffer (50 mM Tris (pH 7.5), 500 mM NaCl, 5 mM EDTA, 1% TritonX-100, 0.1% SDS, 10% Glycerol, 1 mM dithiothreitol) supplemented with protease inhibitor cocktail (Roche), and digested by 450 digital sonifier (Branson). Cell extract were incubated with mouse anti-DDDDK-tag antibody (MBL, M185-3L) and Protein G-conjugated magnetic beads (Invitrogen, 10004D) overnight at 4°C. Samples were washed two times with whole cell lysis buffer and IP buffer (50 mmol/l Tris (pH 7.5), 150 mmol/l NaCl, 1% TritonX-100). Proteins were eluted by incubating with 3x FLAG peptide solution, and analyzed by Western blotting using rabbit anti-GFP antibody (Invitrogen, A11122).

## Live-cell imaging

The *Dnmt1*-KO ESCs stably expressing hKO1-HP1γ was established by the transfection of the linearized CAG- hKO1-HP1γ expressing plasmid. Time-lapse fluorescence microscopy was performed using an upright OLYMPUS confocal microscope (SpinSR10) with an x100 oil immersion lens. Images were taken every 2 min and processed using cellSens (OLYMPUS) and IMARIS (Carl Zeiss) software.

## Hexanediol treatment

To disrupt LLPS, 1,6-hexanediol (Nacalai Tesque, 17913–72) or 2,5-hexanediol (TCI, H0100) at a final concentration of 10% was added to the ESC suspension. After 1 min of incubation at

room temperature, the cells were placed on a glass slide dipped in the fixative. For the recovery experiment, 1,6-hexanediol treated cells were diluted with a 100-fold volume of ESC culture medium and incubated for 1 h in a humidified incubator at 37°C, followed by the fixation on a glass slide.

## Supporting information

**S1 Fig. Enrichment of HP1γ and depletion of H3K9me2 from the chromocenter of female meiotic germ cells and ESCs. (A), (B)**: Representative immunostaining images of surface-spread nuclei at the first meiotic prophase of female. Germ cells were harvested from E15.5-E16.5 embryos. **(C)**: Representative immunostaining images of the wild-type, *Dnmt1*-KO, and *Dnmt1/Tet1*-DKO ESCs. The yellow squares indicate the enlarged areas shown on the right panels. The white dashed circles indicate the chromocenter. **(D)**: Quantification of H3K9me2 enrichment in the chromocenter, related to (C). The signal intensity at the chromocenter was normalized with whole nuclear area. Wild-type, n = 16; *Dnmt1*-KO, n = 16; *Dnmt1/Tet1*-DKO, n = 16. Scale bar, 5 μm.
(TIF)

**S2 Fig. 5hmC deposition at PCH and chromocenter clustering in the DNA hypomethylated cells. (A)**: Representative immunostaining images of surface-spread nuclei of the wild-type, *Dnmt1*-KO and *Dnmt1/Tet1*-DKO ESCs. The yellow squares indicate the enlarged areas shown on the right. The white dashed circles indicate the PCH. **(B), (C)**: Quantification of immunostaining of 5hmC (B) and 5mC (C), related to (A). The enrichment score is calculated by the relative signal intensity of PCH to the entire chromosome. An average of 3 chromosomes was calculated for each cell. n = 20 per cell line. **(D)**: Representative immunostaining images of the wild-type, *Dnmt1*-KO and *Dnmt1/Tet1*-DKO ESCs. The yellow squares indicate the enlarged areas shown on the right. White dashed circles indicate the chromocenter. ESCs were plated in a culture slide and cultured for 1 d before immunostaining. **(E)**: Quantification of TET1 enrichment in the chromocenter, related to (D). The signal intensity at the chromocenter was normalized with whole nuclear area. Wild-type, n = 21; *Dnmt1*-KO, n = 22. **(F)**: Western blotting of whole cell extract from the wild-type, *Dnmt1*-KO, and *Dnmt1/Tet1*-DKO ESCs. **(G)**: Representative immunostaining images of surface-spread nuclei of the wild-type, *Dnmt1/3a/3b*-TKO, and *Dnmt1/3a/3b/Tet1*-QKO ESCs. **(H)**: Boxplot showing the number of distinct chromocenters in the nuclei of ESCs, related to (G). Wild-type, n = 55; *Dnmt1/3a/3b*-TKO, n = 99; *Dnmt1/3a/3b/Tet1*-QKO, n = 55. *P* values were calculated using the Mann-Whitney U-test. *** $P < 0.001$; ** $P < 0.01$; * $P < 0.05$. Scale bar, 5μm.
(TIF)

**S3 Fig. Forced recruitment of TET1-CD induces 5hmC deposition at PCH. (A)**: Quantification of GFP enrichment in the chromocenter, related to Fig 2(C). The signal intensity at the chromocenter was normalized with whole nuclear area. scFv-only, n = 10; dCas9 control, n = 12; TET1-CD, n = 15; TET1-CD-Mut, n = 16. **(B)**: Representative 5hmC immunostaining images of the surface-spread nuclei of the ESCs expressing individual cassette. **(C)**: Quantification of immunostaining of 5hmC, related to (B). scFv-only, n = 14 dCas9 control, n = 11; TET1-CD, n = 14; TET1-CD-Mut, n = 11. *P* values were calculated using the Mann-Whitney U-test. ** $P < 0.01$. Scale bar, 5 μm.
(TIF)

**S4 Fig. *Nanog* expression in the DNA hypomethylated ESCs. (A)**: Representative immunostaining images of the wild-type, *Dnmt1*-KO and *Dnmt1/Tet1*-DKO ESCs. The yellow squares indicate the enlarged areas shown on the right. White dashed circles indicate the

chromocenter. **(B)**: Quantification of NANOG enrichment in the chromocenter, related to (A). The signal intensity at the chromocenter was normalized with whole nuclear area. Wild-type, n = 14; *Dnmt1*-KO, n = 13; *Dnmt1/Tet1*-DKO, n = 15. **(C)**: Representative image of Western blotting analysis of whole cell extracts from the wild-type, *Dnmt1*-KO, and *Dnmt1/Tet1*-DKO ESCs.
(TIF)

**S5 Fig. *Tet1*-dependent PRC2 deposition in the PCH of DNA hypomethylated ESCs. (A)**: Representative immunostaining images of surface-spread nuclei of the wild-type, *Dnmt1*-KO and *Dnmt1/Tet1*-DKO ESCs. The yellow squares indicate the enlarged areas shown on the right. The white dashed circles indicate the PCH. **(B)**: Quantification of immunostaining of H3K27me3, related to (A). n = 20 per cell line. **(C)**: ChIP-qPCR for H3K27me3 at the major satellite repeats in the wild-type, *Dnmt1*-KO and *Dnmt1/Tet1*-DKO ESCs. Each bar represents relative enrichment after normalization against H3 ChIP. n = 3 per cell line. **(D)**: Representative immunostaining images of the wild-type, *Dnmt1*-KO and *Dnmt1/Tet1*-DKO ESCs. The yellow squares indicate the enlarged areas shown on the right. White dashed circles indicate the chromocenter. ESCs were plated in a culture slide and cultured for 1 d before immunostaining. **(E)**: Quantification of EZH2 enrichment in the chromocenter, related to (D). The signal intensity at the chromocenter was normalized with whole nuclear area. Wild-type, n = 15; *Dnmt1*-KO, n = 16; *Dnmt1/Tet1*-DKO, n = 16. *P* values were calculated using the Mann-Whitney U-test. **$P < 0.01$. Scale bar, 5 μm.
(TIF)

**S6 Fig. A minor effect of the PRC2 depletion on chromocenter clustering. (A)**: Schematic illustration showing the designs of sgRNA and genotyping PCR for the depletion of *Ezh2* (top). Results of genotyping PCR confirming the successful deletion of whole *Ezh2* gene locus. **(B)**: Representative immunostaining images of surface-spread nuclei of the *Dnmt1*-KO and *Dnmt1/Ezh2*-DKO ESCs. **(C)**: Representative immunostaining images of the wild-type, *Dnmt1*-KO, and *Dnmt1/Ezh2*-DKO ESCs. The yellow squares indicate the enlarged areas shown on the right. White dashed circles indicate the chromocenter. ESCs were plated in a culture slide and cultured for 1 d before immunostaining. **(D)**: Quantification of TET1 enrichment in the chromocenter, related to (C). The signal intensity at the chromocenter was normalized with whole nuclear area. Wild-type, n = 21; *Dnmt1*-KO, n = 22; *Dnmt1/Ezh2*-DKO #1, n = 19; *Dnmt1/Ezh2*-DKO #2, n = 23. **(E)**: Representative immunostaining images of surface-spread nuclei of the *Dnmt1*-KO and *Dnmt1/Ezh2*-DKO ESCs. **(F)**: Boxplot showing the number of distinct chromocenters in the nuclei of ESCs. *Dnmt1*-KO, n = 55; *Dnmt1/Ezh2*-DKO #1, n = 32; DKO #2, n = 37. **(G)**: Quantification of HP1γ enrichment in the chromocenter, related to (E) and Fig 1(A). The signal intensity at the chromocenter was normalized with whole nuclear area. Wild-type, n = 29; *Dnmt1*-KO, n = 14; *Dnmt1/Tet1*-DKO, n = 18; *Dnmt1/Ezh2*-DKO #1, n = 16; *Dnmt1/Ezh2*-DKO #2, n = 16. *P* values were calculated using the Mann-Whitney U-test. N.S, no significance, $P > 0.05$; **$P < 0.01$. Scale bar, 5 μm.
(TIF)

**S7 Fig. *Tet1*-dependent RING1B localization to chromocenter. (A)**: Representative immunostaining images of the wild-type, *Dnmt1*-KO, *Dnmt1/Tet1*-DKO, and *Dnmt1/Ezh2*-DKO ESCs. Yellow squares indicate the enlarged areas shown on the right. White dashed circles indicate the chromocenter. The ESCs were plated in a culture slide and cultured for 1 d before immunostaining. Scale bar, 5 μm. **(B)**: Quantification of RING1B enrichment in the chromocenter, related to (A). The signal intensity at the chromocenter was normalized with whole nuclear area. Wild-type, n = 19; *Dnmt1*-KO, n = 21; *Dnmt1/Tet1*-DKO, n = 22. **(C)**: Co-

immunoprecipitation analysis of TET1 and RING1B. Cell lysate were immunoprecipitated with anti-IgG or anti-DDDDK-tag (FLAG) antibodies. Western blotting was performed using the antibodies indicated. **(D)**: Representative immunostaining images of surface-spread nuclei of the ESCs expressing scFv-only, dCas9 control, TET1-CD, and TET1-CD-Mut cassette. Yellow squares indicate the enlarged areas shown on the right. White dashed circles indicate the chromocenter. **(E)**: Quantification of RING1B enrichment in the chromocenter, related to (D). The signal intensity at the chromocenter was normalized with whole nuclear area. scFv-only, n = 10; dCas9 control, n = 12; TET1-CD, n = 15; TET1-CD-Mut, n = 16. *P* values were calculated using the Mann-Whitney U-test. $^{**}P < 0.01$. $^{*}P < 0.05$. Scale bar, 5 μm.
(TIF)

**S8 Fig. A minor effect of the PRC2 depletion on RING1B localization to chromocenter.**
**(A)**: Representative immunostaining images of the wild-type, *Dnmt1*-KO, *Dnmt1/Tet1*-DKO, and *Dnmt1/Ezh2*-DKO ESCs. Yellow squares indicate the enlarged areas shown on the right. White dashed circles indicate the chromocenter. The ESCs were plated in a culture slide and cultured for 1 d before immunostaining. Scale bar, 5 μm. **(B)**: Quantification of RING1B enrichment in the chromocenter, related to (A). The signal intensity at the chromocenter was normalized with whole nuclear area. Wild-type, n = 19; *Dnmt1*-KO, n = 21; *Dnmt1/Tet1*-DKO, n = 22; *Dnmt1/Ezh2*-DKO #1, n = 22; *Dnmt1/Ezh2*-DKO #2, n = 25. **(C)**: Representative immunostaining images of surface-spread nuclei of the wild-type, *Dnmt1*-KO and *Dnmt1/Ezh2*-DKO ESCs. The yellow squares indicate the enlarged areas shown on the right. The white dashed circles indicate the PCH. **(D)**: Quantification of immunostaining of H2AK119ub, related to (C). n = 20 per cell line. *P* values were calculated using the Mann-Whitney U-test. $^{**}P < 0.01$. Scale bar, 5 μm.
(TIF)

**S9 Fig. TET1 localization to chromocenter of DNA hypomethylated ESCs was partly regulated by PRC1. (A)**: Schematic illustration showing the designs of sgRNA for the depletion of *Ring1A* (top) and *Ring1B* (bottom). **(B)**: Representative immunostaining images of surface-spread nuclei of the wild-type, *Dnmt1*-KO and *Dnmt1/Ring1A/1B*-TKO ESCs. Cells were analyzed 4 d after the transfection of the CRISPR cassette. **(C)**: Quantification of TET1 enrichment in the chromocenter, related to (B). The signal intensity at the chromocenter was normalized with whole nuclear area. Wild-type, n = 20; *Dnmt1*-KO, n = 22; *Dnmt1/Ring1A/1B*-TKO, n = 22. *P* values were calculated using the Mann-Whitney U-test. $^{**}P < 0.01$. Scale bar, 5 μm.
(TIF)

**S10 Fig. Effect of 1,6-hexanediol (HD) treatment on the chromocenter architecture in the wild-type ESCs. (A)**: Representative immunostaining images of surface-spread nuclei of the wild-type ESCs with or without HD treatment. About 10% of 2,5- or 1,6-HD were treated for 1 min. **(B)**: Boxplot showing the number of distinct chromocenters in the nuclei of wild-type ESCs treated with 2,5- or 1,6-HD, related to (A). No treatment, n = 111; 2,5-HD treatment, n = 179; 1,6-HD treatment, n = 105. *P* values were calculated using the Mann-Whitney U-test. N.S., no significant, $P > 0.05$. $^{*}P < 0.05$.
(TIF)

**S1 Movie. Time-lapse confocal microscopy imaging of the *Dnmt1*-KO ESCs with the hKO1-HP1γ reporter.**
(MP4)

**S1 Table. sgRNA sequences.**
(DOCX)

## Acknowledgments

We thank A. Masumoto for helping with this work. We thank M. Okano for *Dnmt1*-KO and *Dnmt1/3a/3b*-TKO ESCs. A part of microscopy analysis of this study was supported by the Center for Medical Research and Education, Graduate School of Medicine, Osaka University.

## Author Contributions

**Conceptualization:** Yota Hagihara, Shinpei Yamaguchi.

**Data curation:** Yota Hagihara, Shinpei Yamaguchi.

**Investigation:** Yota Hagihara, Satoshi Asada, Takahiro Maeda, Shinpei Yamaguchi.

**Methodology:** Yota Hagihara.

**Supervision:** Toru Nakano, Shinpei Yamaguchi.

**Writing – original draft:** Yota Hagihara.

**Writing – review & editing:** Toru Nakano, Shinpei Yamaguchi.

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
