## [Decision Letter · Decision Letter 0]

29 Mar 2021

Dear Dr Yamaguchi,

Thank you very much for submitting your Research Article entitled 'Tet1 regulates epigenetic remodeling of the pericentromeric heterochromatin and chromocenter organization in DNA hypomethylated cells' to PLOS Genetics.

The manuscript was fully evaluated at the editorial level and by two independent peer reviewers. The reviewers appreciated the attention to an important problem, but raised several concerns about the current manuscript. Based on the reviews, we will not be able to accept this version of the manuscript, but we would be willing to review a revised version. We cannot, of course, promise publication at that time.

Should you decide to revise the manuscript for further consideration here, your revisions should address each of the specific points made by the reviewers. The revised manuscript must include a detailed list of your responses to the review comments and a description of the changes you have made. In particular, we expect that a revised manuscript should:

1. address the role of the catalytic activity of TET1 in mediating chromocenter clustering through the interaction between TET1 and PRC1 complex;

2. elaborate on the involvement of PRC1/RING1B on DAPI dense regions in forced TET1 targeting experiments;

3. clearly describe quantification and statistical methods using in modified histone immunostaining experiments (i.e. H2AK119Ub and H3K27me3) and provide verification of enrichment by another method, such as ChIP. if possible.

If you decide to revise the manuscript for further consideration at PLOS Genetics, please aim to resubmit within the next 60 days, unless it will take extra time to address the concerns of the reviewers, in which case we would appreciate an expected resubmission date by email to plosgenetics@plos.org.

[LINK]

We are sorry that we cannot be more positive about your manuscript at this stage. Please do not hesitate to contact us if you have any concerns or questions.

Yours sincerely,

Beth A. Sullivan, PhD

Associate Editor

PLOS Genetics

John Greally

Section Editor: Epigenetics

PLOS Genetics

Reviewer's Responses to Questions

**Comments to the Authors:**

Reviewer #1: Hagihara et al (PGENETICS-D-21-00261)

In this study, the authors described molecular mechanisms of chromocenter clustering at pericentromeric heterochromatin (PCH) using Dnmt1 KO mouse ES cell (mESC) as a model for DNA hypomethylated female meiotic germ cell. Dnmt1 KO DNA hypomethylated mESCs show 5hmC enrichment on PCH and chromocenter clustering (decreased number of DAPI dense foci and HP1g accumulation on them) like female meiotic germ cells. Also, clustered chromocenters in Dnmt1 KO mESCs are enriched for H3K27me3, H2AK119ub, RING1B and TET1. The authors demonstrated that the chromocenter clustering and associating epigenetic changes (5hmC, H3K27me3 and H2AK119ub) are TET1 dependent. Furthermore, the TET1-mediated chromocenter clustering is PRC1 dependent. Finally, the authors showed that the chromocenter clustering is sensitive to 1,6-hexanediol (HD), suggesting that LLPS is involved in this formation in the DNA hypomethylated condition.

Their findings are interesting and provided data are mostly solid. However, if this work focuses on the molecular mechanism of chromocenter clustering in the hypomethylated cells, especially the role of TET1 and PRCs, some of issues remain to be clarified yet. Furthermore, some of different biochemical data should be included for validating their proposal/interpretation. Please tackle following reviewer’s comments in order to improve this manuscript as more complete work.

Major comments

1. Fig. 1A and 3C

a) “The condensed chromocenter of the Dnmt1-KO ESCs retained a strong enrichment of HP1γ, which is similar to meiotic germ cells (Fig. 1A and S1A).” line 125-126.

This finding is interesting. Is this H3K9me3 dependent? Please clarify this. Although this is not a main issue of chromocenter clustering mechanisms, the HP1g enrichment seems to be epigenetically regulated. At least, Fig. 3C data indicates that this phenomenon is chromocenter clustering (compaction?) independent.

b) One more, "HP1γ enrichment at PCH did not change in the Dnmt1/Ring1A/1B-TKO ESCs, suggesting that HP1γ was dispensable for chromocenter clustering under DNA hypomethylated condition. line 213-215.

“HP1g enrichment or accumulation was dispensable” is OK, but “HP1g was dispensable” is say too much. This sentence should be revised.

c) Tet1-KO should be demonstrated by TET1 Ab Western blot analysis.

2. Fig. 2 and S2E, F

a) To further validate the dCas9-GCN4/scFvGFP forced recruitment system, the authors should show specific GFP signals of scFv/GFP/Tet1-CD on PCH only at the condition of major satellite repeat sgRNA+dCas9-GCN4 but not at scFv-only or dCas9 control as shown in Fig. 2B and C.

b) “To investigate whether the chromocenter clustering is mediated by the catalytic activity dependent or -independent function of TET1,” line 152-153.

If the authors challenge this problem, the reviewer suggests performing the experiment (Fig. 2) using forced recruitment of TET1-mutantCD (catalytic dead).

c) The results of Dnmt1/3a/3b-TKO and Dnmt1/3a/3b/Tet1-QKO ESCs (Fig. S2E and F) are very clear, but at least the statement “the catalytic activity of TET1 are dispensable for chromocenter clustering.” line 160-161 should be revised because these results do not specifically address the catalytic activity of TET1.

d) One more, if essential region of PRC1 (RING1B) interaction in TET1 CD is known, the experiment of forced recruitment of the RING1B interaction defective mutant TET1-CD is also recommended in order to validate whether PRC1 interaction is essential for the forced TET1-CD recruitment-mediated chromocenter clustering. Furthermore, examination of the enrichment of PRC1 (RING1B) on DAPI dense regions by forced recruitment of TET1 CD (WT) in WT ESCs is another suggesting experiment.

3. Fig. 3A

H2AK119ub enrichment on DAPI dense regions in Dnmt1 KO ESCs should be validated by H2AK119ub ChIP-qPCR (major satellite repeat). And, Tet1 KO antagonizes this phenotype?

4. Fig. 4B

If Dnmt1 KO specific dynamics of chromocenter rearrangement by 1,6-HD can be observed, it is very nice. If possible, try time-lapse fluorescence microscopic analysis of Hoechst33342 dense foci after the treatment of 1,6-HD (during 1 min). Hoechst33342 can be used for live cell imaging.

5. Figure 5 Functional relationship among TEF1, PRC1 and PRC2

Should address following issues.

a) TET1 is essential for PRC1 accumulation on PCH in Dnmt1 KO ESCs, but how about TET1 enrichment on PCH in Dnmt1/Ring1A/1B-TKO ESCs?

b) How about PRC1, H2AK119ub, TET1 and HP1� enrichment on PCH in Dnmt1/Ezh2-DKO ESCs?

These results further clarify functional relationship among them for chromocenter clustering.

6. Fig S4A

The Tet1 and Dnmt1-KO dependent H3K27me3 PCH enrichment should be validated by H3K27me3 ChIP-qPCR. Also, examine EZH2 enrichment on them, at least by immunostaining.

7. Fig. S5A

The Tet1 and Dnmt1-KO dependent RING1B enrichment on DAPI dense regions should be quantitatively measured and validated statistically.

8. Fig. S6

Control data (no HD treated WT cell) should be included.

Minor comments,

9. The authors should discuss (at least one sentence) whether the NANOG pathway interacts with the TET1-PRC1 pathway for chromocenter clustering in mESCs.

10. hKO1-HP1γ

Need more explanation. What is hKO1.

11. “The CXXC domain, which can bind to unmethylated DNA, is present in TET1 but not in TET2, and this difference may account for the difference in PCH targeting.” line 257-259.

The authors should cite original work if it is already known that the CXXC domain of TET1 binds to unmethylated DNA.

12. P14 line 268. Fig. S3A, S3B→Fig. S4A, S4B?

Reviewer #2: How histone H3K27me3 compensates the loss of H3K9me2/3 or DNA methylation at pericentromeric heterochromatin (PCH) has puzzled this field for a long time. In this manuscript, Hagihara et al. showed mostly by immunostaining that TET1 plays a central role in the recruitment of polycomb repressive complexes to reorganizing chromocenters in Dnmt1 knockout mouse embryonic stem cells. First, they revealed that TET1-depletion reverted the clustered chromocenters in Dnmt1 KO mouse ES cells to scattered smaller ones. Then, they validated the effect of TET1 by ectopically tethering the catalytic domain of TET1 to PCH and found that forced recruitment of TET1-CD was sufficient to induce chromocenter clustering. PRC1 and PRC2 have been shown to localize to PCH in response to DNA hypomethylation. The authors further showed that PRC1 was critical for chromocenter clustering and the deposition of H2AK119ub was dependent on the presence of TET1. Finally, with time-lapse imaging and chemical perturbation, the authors found that the chromocenter clustering in Dnmt1 KO mouse ES cells was driven by liquid-liquid phase separation.

This manuscript provided very clear results with high quality. The demonstration for the essential role of TET1 in mediating chromocenter clustering in Dnmt1 KO cells is convincing. The authors proposed that the underlying mechanism is through the interaction between TET1 and PRC1 complex and indispensable on the enzymatic activity of TET1. However, there is no direct evidence against the involvement of TET1's catalytic activity in the manuscript. Tethering a catalytic-dead form of TET1-CD to PCH to examine its effect on chromocenter clustering will help to clarify this.

Another major concern is that the authors tried to associate the observations on the function of TET1 in Dnmt1 KO cells with that in meiotic female germ cells regardless of inconsistent features. In their previous publication (Nature, 2012), they revealed that Tet1 deficiency resulted in no significant change in HP1γ staining in the oocyte. The current study, on the contrary, showed in the first main figure that Tet1 deletion diminished HP1γ signals at DAPI-dense chromocenters in Dnmt1 KO cells, suggesting the PCH organization mechanism in DNA hypomethylated ES cells is quite different from that in female germ cells. The authors need to revise their manuscript and reconsider their statements related to the phenomenon in female meiosis. In addition, the authors showed that PCH in female germ cells was devoid of H3K9me2 in Fig. S1B. Is it also true in Dnmt1 KO ES cells?

Minor concern:

In Lines 214 and 215, the authors concluded that "HP1γ was dispensable for chromocenter clustering under DNA hypomethylated condition." From Fig. 3C, HP1γ still localized to DAPI-dense chromocenters in Dnmt1/Ring1a/Ring1b triple knockout cells and the chromocenter clustering was ameliorated. This result only indicated that HP1γ was insufficient for chromocenter clustering rather than dispensable.

**Have all data underlying the figures and results presented in the manuscript been provided?**

Reviewer #1: Yes

Reviewer #2: Yes

PLOS authors have the option to publish the peer review history of their article (what does this mean?). If published, this will include your full peer review and any attached files.

Reviewer #1: No

Reviewer #2: **Yes: **Bing Zhu

---

## [Editor Report · Decision Letter 1]

4 Jun 2021

Dear Dr Yamaguchi,

Thank you for submitting the revised version of your manuscript entitled "Tet1 regulates epigenetic remodeling of the pericentromeric heterochromatin and chromocenter organization in DNA hypomethylated cells". We appreciate that you thoughtfully addressed all the review comments and included important new data using the catalytically inactive TET1 targeting. We are pleased to inform you that your manuscript has been editorially accepted for publication in PLOS Genetics. Congratulations!

Yours sincerely,

Beth A. Sullivan, PhD

Associate Editor

PLOS Genetics

John Greally

Section Editor: Epigenetics

PLOS Genetics

Comments from the reviewers (if applicable):

**Data Deposition**

http://datadryad.org/submit?journalID=pgenetics&manu=PGENETICS-D-21-00261R1

**Press Queries**

---

## [Editor Report · Acceptance letter]

18 Jun 2021

PGENETICS-D-21-00261R1 

Tet1 regulates epigenetic remodeling of the pericentromeric heterochromatin and chromocenter organization in DNA hypomethylated cells 

Dear Dr Yamaguchi, 

We are pleased to inform you that your manuscript entitled "Tet1 regulates epigenetic remodeling of the pericentromeric heterochromatin and chromocenter organization in DNA hypomethylated cells" has been formally accepted for publication in PLOS Genetics! Your manuscript is now with our production department and you will be notified of the publication date in due course.

With kind regards,

Katalin Szabo

PLOS Genetics

On behalf of:
